# Depletion of Embryonic Macrophages Leads to a Reduction in Angiogenesis in the Ex Ovo Chick Chorioallantoic Membrane Assay

**DOI:** 10.3390/cells10010005

**Published:** 2020-12-22

**Authors:** Hanna Tay, Charis Du Cheyne, Kristel Demeyere, Jurgen De Craene, Lobke De Bels, Evelyne Meyer, Andries Zijlstra, Ward De Spiegelaere

**Affiliations:** 1Department of Morphology, Ghent University, 9820 Merelbeke, Belgium; Hanna.Tay@UGent.be (H.T.); Charis.DuCheyne@UGent.be (C.D.C.); Jurgen.Decraene@UGent.be (J.D.C.); Lobke.Debels@UGent.be (L.D.B.); 2Cancer Research Institute Ghent (CRIG), Ghent University, 9820 Merelbeke, Belgium; 3Department of Pharmacology, Toxicology and Biochemistry, Ghent University, 9820 Merelbeke, Belgium; Kristel.Demeyere@UGent.be (K.D.); Evelyne.Meyer@UGent.be (E.M.); 4Department of Pathology, Microbiology and Immunology, Vanderbilt University Medical Center, Nashville, TN 37232, USA; andries.zijlstra@vumc.org

**Keywords:** macrophages, macrophage depletion, clodronate liposomes, angiogenesis, ex ovo CAM assay

## Abstract

Macrophages play an important but poorly understood role in angiogenesis. To investigate their role in vessel formation, relevant in vivo models are crucial. Although the chick chorioallantoic membrane (CAM) model has been frequently used as an angiogenesis assay, limited data are available on the involvement of chicken macrophages in this process. Here, we describe a method to deplete macrophages in the ex ovo chick CAM assay by injection of clodronate liposomes and show that this depletion directly affects vascularisation of collagen onplants. Chicken embryos were injected intravenously with either clodronate or phosphate-buffered saline (PBS) liposomes, followed by placement of collagen type I plugs on the CAM to quantify angiogenic ingrowth. Clodronate liposome injection led to a significant 3.4-fold reduction of macrophages compared with control embryos as measured by immunohistochemistry and flow cytometry. Furthermore, analysis of vessel ingrowth into the collagen plugs revealed a significantly lower angiogenic response in macrophage-depleted embryos compared with control embryos, indicating that chicken embryonic macrophages play an essential function in the development of blood vessels. These results demonstrate that the chick CAM assay provides a promising model to investigate the role of macrophages in angiogenesis.

## 1. Introduction

Macrophages are best known as phagocytic cells that play a crucial role in the body’s immune response. They act as antigen-presenting cells, produce inflammatory and antimicrobial cytokines and phagocytise bacteria and foreign objects. However, macrophages possess a high plasticity and also regulate processes such as angiogenesis, the development of new blood vessels from existing vasculature [1].

Angiogenesis is a complex multistep process that is controlled mainly by cells within the tissue to be vascularised such as pericytes, immune cells or tumour cells [2,3]. Accumulating evidence suggests that macrophages play an important but still not fully understood function during all the steps of the angiogenic cascade [3]. To study the involvement of macrophages in angiogenesis, relevant in vitro and in vivo models are required [4]. Although in vitro models have the advantage of limited costs, ease of standardisation, manipulation and real-time imaging, they lack functional blood flow and the complex environment in which blood vessels naturally grow. In vivo models, such as mice models, are biologically more relevant but come at a higher cost, and real-time imaging as well as manipulations in these models are relatively complex [5]. Besides the technical and practical problems in many in vivo models, ethical issues remain a point of discussion when laboratory animals are involved [6].

The chick embryo chorioallantoic membrane (CAM) assay is an exceptional in vivo model, as it possesses some of the advantages of in vitro models, such as low cost, ease of manipulation and real-time imaging [7]. The CAM is a highly vascular yet not innervated membrane which results from the fusion of two extra-embryonic membranes, i.e., the chorion and the allantois, starting at embryonic Day 5 [8]. Between Day 5 and Day 12, the CAM’s vascular network grows in size and complexity, initially through sprouting angiogenesis but, from Day 8, mainly through intussusceptive angiogenesis. Between Day 13 and 14, the blood vessels establish their mature morphology [9]. The CAM model has been widely used since its first introduction as an angiogenesis assay in the 1970s, and this continuing popularity can be attributed to several factors [7,10]. First, fertilised chick eggs are a cost-effective vertebrate system generated through a large-scale commercial enterprise from a homogenous stock. The inbred but robust nature of broiler chickens ensures a stable supply of homogenous eggs that support reproducible experimentation. Second, the model is cost-effective because the egg is self-sustainable. Moreover, very few resources are required to keep the embryo’s in culture under standardized conditions. Third, the short developmental time of the chick embryo (20–21 days) allows for a fast screening of different testing conditions. Fourth, the superficial location of the CAM directly underneath the eggshell makes its manipulation easy. When the embryo is transferred to a recipient and grown ex ovo, the CAM develops on the surface, making it even more accessible for manipulations and for real-time imaging at high resolution [11]. Fifth, as the chick embryo only develops an adequate immune system at Day 15 post-fertilisation, this model is suitable to study angiogenesis in xenograft onplants or after seeding of xenogeneic cells such as macrophages [12,13].

Studies in mice and zebrafish revealed that macrophages play a role at almost each step of the angiogenic cascade, going from the budding of vessel sprouts to vessel maturation [3]. They have been acknowledged as paracrine regulators of angiogenesis, as they are able to secrete multiple tissue-remodelling and pro-angiogenic cytokines such as matrix metalloproteinase (MMP), vascular endothelial growth factor A (VEGFA), angiotropin and fibroblast growth factor 2 (FGF2) [1,14]. Furthermore, macrophages can act as cellular chaperones that physically interact with endothelial tip cells and facilitate the anastomosis of vascular sprouts [15]. Additionally, it was demonstrated that embryonic macrophages play a role in developmental angiogenesis in mice and zebrafish [15,16]. Despite the importance of macrophages in (embryonic) angiogenesis, data about their function in the CAM model are scarce. Monocytes/macrophages have been shown to be a major source of MMP13 in chick embryos, and evidence suggests that MMP13 is the endogenous collagenase that is crucial for matrix remodelling in the CAM and even for induction of angiogenesis [17]. Otherwise, chicken embryonic macrophages are mostly known for phagocytising dead cells and debris and participating in tissue remodelling [18]. To further enhance the potential of the CAM assay as angiogenesis model, we aimed to investigate the involvement of chicken embryonic macrophages in blood vessel development by means of macrophage depletion.

In the absence of genetically modified chicken lines for targeted macrophage depletion, this depletion can easily be achieved using clodronate liposomes [19]. Clodronate is a biphosphonate which metabolises intracellularly to a cytotoxic variant of ATP, disrupting the cell’s energy metabolism and leading to cell death [20]. Clodronate-containing liposomes are ingested by phagocytes and, more specifically, macrophages, causing a general depletion of the macrophage population without affecting other cell types and without the need for targeted knockdown of specific genes [19]. Macrophage depletion after treatment with clodronate liposomes has been widely used in macrophage research, for example, in research concerning the so-called tumour-associated macrophages (TAMs), where a reduction in the vascularisation and subsequently size of the investigated tumour could be detected after macrophage depletion [21,22]. Another example is the use in larval zebrafish to research the role of peripheral macrophages in tissue regeneration [23]. In the present study, we investigated whether clodronate liposomes can also be used in the chick CAM assay for targeted depletion of macrophages. Subsequently, we assessed whether clodronate liposome-mediated macrophage depletion affects angiogenesis in collagen onplants on the CAM.

## 2. Materials and Methods

### 2.1. Ex ovo CAM assay

According to the European Directive 2010/63/EU, avian embryos are not specified as laboratory animals. Therefore, no approval of the local Ethics Committee of the Faculty of Veterinary Medicine and the Faculty of Bioscience Engineering of Ghent University was required. Fertilised eggs from broiler chickens (Ross 308), pre-incubated for 60 to 72 h, were obtained at a local hatchery (Broeierij David, Tielt, Belgium). One lateral side of the eggshells was disinfected with povidone iodine and horizontally incubated with the disinfected side facing downwards at 37.8 °C for 15 min to allow the embryo to position itself on top of the yolk. Subsequently, small indents were made in the shell using a cut-off wheel (Dremel). The egg content was transferred to square weighing boats (86 × 86 × 25 mm, Z154881-500EA, Sigma-Aldrich, Darmstadt, Germany) (Figure 1).

The embryos were incubated at 37.8 °C and 80% relative humidity, and the cover of a square Petri dish (SIMPD210-16, VWR International, Leuven, Belgium) was used as a lid.

### 2.2. Depletion of Macrophages by Clodronate Liposome Injection

After 8.5 days of incubation, embryos were injected intravenously in the CAM with 100 µL, 50 µL or 25 µL of 5 mg/mL clodronate liposomes or phosphate-buffered saline (PBS) liposomes as a control (LIPOSOMA Research Liposomes, Amsterdam, The Netherlands). According to the manufacturer, the size of both clodronate and PBS liposomes measures 1.7 µm on average, with a minimum of 150 nm and a maximum of 3 µm. As there was no difference in depletive effect between the injected volumes (Appendix A) and as the manufacturer states that the recommended volume for intravenous injection of clodronate liposomes is 1 mL per 100 g of animal weight and chicken embryos weigh around 2.5 g after 8–9 days of incubation, we opted to inject 25 µL of liposomes going forward. The intravenous injection was carried out using a pulled capillary needle and a 1 cm^3^ syringe, connected to each other using a tight-fitting tube (Figure 1). During injection, the CAM was pulled back with a cotton swab to allow the needle to penetrate the blood vessel. To quantify macrophage depletion, a number of embryos were euthanised 1 to 3 days after injection using 200 µL of 400 mg/mL natriumpentobarbital (KELA, Hoogstraten, Belgium) administered intravenously. To confirm that liposomes were only ingested by macrophages as stated by the manufacturer, fluorescent Dil liposomes (I-005, LIPOSOMA Research Liposomes, Amsterdam, The Netherlands) were injected in a similar way, after which embryos were recuperated for flow cytometry.

### 2.3. Immunohistochemistry

Embryos (*n* = 16; 8 control, 8 clodronate liposome-injected) were embedded in Frozen Section Medium (Richard-Allan Scientific Neg-50 6502, Thermo Fisher Scientific, Merelbeke, Belgium) and snap-frozen in isopropanol cooled by liquid nitrogen. Frozen sections were cut at 7 µm thickness and collected on HistoBond slides (0810001, Paul Marienfeld GmbH & Co. KG, Lauda-Königshofen, Germany). The sections were post-fixated in ice-cold acetone for 2 min. To avoid non-specific staining, slides were treated with 30% rabbit serum for 30 min. Chicken macrophages were detected using a 1/800 dilution of mouse anti-chicken primary antibody, clone KUL01 (Cat# 8420-09, Southern Biotech, Birmingham, AL, USA) for 1 h at room temperature (RT). Endogenous peroxidase activity was quenched by incubation with 0.03% hydrogen peroxide for 5 min. Subsequently, sections were treated with Dako EnVision+ System-HRP Labelled Polymer Anti-mouse (#K4001, Agilent, Santa Clara, CA, USA) for 30 min and Dako Liquid DAB+ Substrate Chromogen System (#K3468, Agilent, Santa Clara, CA, USA) for 5 min at RT. Slides were counterstained with haematoxylin, dehydrated in a graded series of alcohol and mounted with dibutylphthalate polystyrene xylene (DPXImageJ) (Appendix A). The number of crossing points of the grid with embryonic tissue was determined, as well as the number of positive cells. The ratio of positive cells per crossing point was then calculated.

### 2.4. Flow Cytometry

For flow cytometry, embryos (*n* = 12; 6 control, 6 clodronate liposome-injected) were minced in pieces of approximately 1 mm with a fine scalpel and further digested in 5 mL Dispase II (SCM133, Merck Millipore, Darmstadt, Germany) at 37 °C while shaking. After 25 min, an equal amount of PBS with 5 mM ethylenediaminetetraacetic acid (EDTA) and 2% heat-inactivated foetal bovine serum (FBS) was added to block Dispase II activity. This solution was filtered through a 40 µm cell strainer. The filtered solution was centrifuged at 300× *g* for 10 min and the cell pellet was washed twice with 1 mM PBS-EDTA and spun at 300× *g* for 5 min. To enrich the total amount of leucocytes, the cell suspension was incubated in a 1/500 dilution of CD45-APC (allophycocyanin) mouse anti-chicken leucocyte marker (Cat# 8270-11, Southern Biotech, Birmingham, AL, USA) for 15 min in the dark at RT, washed twice then incubated in a 1/200 dilution of secondary biotinylated antibody (E0464, Agilent, Santa Clara, CA, USA) for 30 min at RT, and washed twice again. Next, CD45-positive cells were enriched using the EasySep Biotin Positive Selection Kit II (#17683, Stem Cell Technologies, Grenoble, France) and EasySep Magnet (#18000, Stem Cell Technologies, Grenoble, France), using the manufacturer’s protocol. In short, 100 µL of the biotin selection cocktail was added per ml cell suspension and incubated for 15 min at RT. The magnetic particles were mixed and 50 µL per 1 mL sample was added and incubated for 10 min at RT. The sample was then topped up to 2.5 mL with 1 mM PBS-EDTA supplemented with 2% FBS, placed into the magnet and incubated at RT for 5 min, after which the supernatant was removed. The incubation step in the magnet was repeated twice to maximise the purity of the isolated CD45-positive cells.

Staining of chicken macrophages with (phycoerythrin) PE-conjugated KUL01 antibody (Cat# 8420-09, Southern Biotech, Birmingham, AL, USA) was performed in the enriched leucocyte population. The Fc region blocking was done using 10% heat-inactivated chicken serum for 10 min at RT. After 1 wash step, the cell pellet was resuspended in 1/200 diluted primary antibody and incubated for 15 min at RT in the dark. An isotype control was performed using PE-labelled Mouse IgG1k (#12-4714-41, eBioscience, San Diego, CA, USA) at the same 1/200 dilution (Appendix A). The cell suspensions were washed twice and acquired with the CytoFLEX (Beckman Coulter) and subsequently analysed with CytExpert v2.0 software. Cell viability was checked with propidium iodide 1/50 diluted in an unstained sample (Appendix A). After gating to exclude dead cells and debris, the percentage of KUL01-PE positive macrophages was determined in the embryos’ CD45-positive leucocyte population.

Staining the chicken macrophages after injection with fluorescent Dil liposomes was performed in exactly the same way, with the only difference being that the anti-macrophage antibody KUL01 was conjugated with FITC (fluorescein isothiocyanate) (Cat# 8420-02, Southern Biotech, Birmingham, AL, USA) instead of PE, as the emission spectra of Dil closely resembles that of PE.

### 2.5. CAM Angiogenesis Assay

The angiogenic effect of macrophage depletion on the CAM was analysed using a method described by Zijlstra and colleagues [24]. Collagen Type I (REF 354249, High Concentration, Corning, Bedford, MA, USA) was used to make collagen plugs with an end concentration of 2 mg/mL. Both plain and angiogenic collagen plugs were made. Angiogenic plugs contained 5 × 10^4^ human fibrosarcoma (HT1080) cells per plug, which were previously shown to induce angiogenesis [24]. Plain collagen plugs did not contain cells. The plugs were placed on a mesh to allow quantification of angiogenesis as described by Nguyen and colleagues [25]. The use of collagen/mesh onplants for measurement of angiogenesis guaranteed that only newly developed blood vessels would be counted during angiogenic scoring, as the new vessels were forced to grow vertically in the plug and were therefore easily distinguishable from pre-existing vessels in the horizontal plane of the CAM.

Collagen plugs were made by first neutralising high-concentration collagen Type I to a pH of 7.4 by adding a mixture of 10× PBS, 1 M HEPES, 1 N NaOH and sterile water at 4 °C. The final concentration of collagen in this mixture was 3 mg/mL. This ice-cold neutralised collagen was combined with either a mixture of 1× PBS and 20 × 10^6^ HT1080 cells/mL in a cell medium to produce angiogenic plugs (resulting in 5 × 10^4^ cells per 30 µL plug), or a mixture of 1× PBS and the cell medium for plain collagen plugs. After pipetting up and down at least 10 times to ensure thorough mixing of the collagen without altering its consistency, droplets of 30 µL were pipetted onto two layers of sterilised nylon mesh (bottom mesh, 4 × 4 mm; top mesh, 2 × 2 mm; 03-150/38 Sefar Nitex, Sefar B.V., Lochem, The Netherlands), which were precoated with 0.2% Pluronic (Ref: P2443-250G, Sigma-Aldrich, Darmstadt, Germany) to avoid adhesion of cells to the mesh. The collagen/mesh onplants were left to polymerise at 37°C for 1 h.

Collagen/mesh onplants were placed on top of the CAM of 8.5-day-old embryos immediately after intravenous injection of clodronate liposomes (*n* = 26) or PBS liposomes (*n* = 32) (Figure 1). Each embryo received two angiogenic plugs and two plain collagen plugs. After 3 days, images of the plugs were taken with a fluorescence stereo zoom microscope (Axio.Zoom V16, Zeiss) in the bright field and DAPI (4’,6-diamidino-2-phenylindole) (excitation, 358 nm; emission, 463 nm) channels to ensure proper visualisation of the blood vessels. Images were analysed with the software ZEN Blue v2.6. and Fiji (is just ImageJ). In the DAPI channel, blood vessels growing in the collagen plug above the plane of the top mesh could be identified as black structures on a grey background. Bright-field images were used when the DAPI image did not provide enough certainty. All blood vessels growing above the plane of the top mesh were counted. This was performed in a blinded fashion to avoid bias. The angiogenic response of each plug was calculated as the percentage of squares in the top mesh that contained blood vessels (=amount of positive squares/total squares). The choice to perform the CAM angiogenesis assay between Days 8 and 12 of embryonic development was guided by the fact that the CAM’s vascular network grows mainly between Days 5 and 12 [9], and that placement of collagen plugs requires a sufficiently large CAM which is only achieved around Days 8–9.

To confirm whether macrophage depletion was also detected at the level of the collagen/mesh onplants, a number of onplants of both PBS liposome and clodronate liposome-injected embryos were recuperated for immunohistochemistry.

To confirm that the angiogenic effect was due to the depletion of macrophages and not caused by possible free clodronate released from liposomes, free clodronate (ab141919, abcam, Cambridge, UK) was injected at the same concentration and volume as the liposome-encapsulated clodronate (5 mg/mL in a volume of 25 µL) (*n* = 18). Control embryos were injected with PBS (*n* = 17). The CAM angiogenesis assay was then performed as described above.

### 2.6. Embryo Viability after Injection with Clodronate Liposomes and Free Clodronate

For each experiment performed with the CAM angiogenesis assay (*n* = 8), embryos were divided into two equal groups and injected with either PBS liposomes or clodronate liposomes, or PBS/free clodronate. Aside from taking images of the collagen plugs 3 days after injection, the number of living embryos in each group was also recorded to determine whether injection with clodronate liposomes or free clodronate had an impact on embryo viability.

### 2.7. Statistics

Statistical analysis was performed with the software R Project for Statistical Computing v3.6.3. The Wilcoxon rank sum test was used to analyse differences between groups. The Cochran–Mantel–Haenszel Test was used to analyse embryo viability. The R packages ggplot2 (version 3.3.0) and ggpubr (version 0.2.5) were used for data visualisation.

## 3. Results

### 3.1. Clodronate Liposome Injection Depletes Macrophages in the Chicken Embryo

The effect of clodronate liposome treatment on endogenous macrophages was evaluated in the developing choroid and the mesenchyme surrounding the sclera of the chick eye. These are major sites of macrophage infiltration because of their importance in vascular development of the eye and central nervous system [18]. Immunohistochemical analysis using the chicken macrophage-specific marker KUL01 revealed a reduction of the KUL01-positive cells from the first day after injection of clodronate liposomes until at least 3 days after injection (Figure 2a). On the third day after injection, a 3.4-fold decrease in the median count of KUL01-positive cells was observed in clodronate liposome-treated embryos compared with control embryos injected with PBS liposomes (** *p* < 0.001) (Figure 3a; Appendix A). Collagen/mesh onplants of clodronate liposome-injected embryos collected at 3 days after onplanting contained some but notably fewer macrophages compared with onplants on PBS liposome-injected embryos (Appendix A).

Dual-staining flow cytometry confirmed the effect of clodronate liposome injection on the macrophage population. In control embryos, the KUL01-positive cells ranged between 15–37% of CD45-positive leucocytes, while this amount decreased to 3–9% in clodronate liposomes-injected embryos (Figure 2b). Furthermore, flow cytometry of embryos injected with Dil liposomes, which are also taken up by phagocytosis but are not cytotoxic, confirmed that liposomes are mainly ingested by macrophages, as around 90% of the cells which accumulated Dil liposomes were also positive for FITC-conjugated KUL01 (Appendix A).

### 3.2. Injection of Clodronate Liposomes but not Free Clodronate Reduces Viability of the Chicken Embryo

To estimate the effect on the overall survival of clodronate liposome injection in embryos, we compared the embryo survival rate 3 days after injection of clodronate or PBS liposomes. Across all experiments, 56.0% of clodronate liposome-injected embryos survived versus 82.5% of PBS liposome-injected embryos (Figure 3b; Appendix A), indicating a significantly lower survival rate in chicken embryos injected with clodronate liposomes (* *p* = 0.01988). To confirm that the decreased viability was due to the depletion of the macrophages and not the free clodronate released from the liposomes, we also tested the viability in embryos in which free clodronate was injected at the same concentration as in the clodronate liposome-treated embryos. Embryo viability after injection with free clodronate or PBS showed no significant difference in survival rate (*p* = 0.8176). Across all experiments, 80.56% of embryos injected with free clodronate survived versus 88.57% of PBS-injected embryos (Appendix A).

### 3.3. Depletion of Macrophages Induces a Reduced Angiogenic Response

To allow quantification of angiogenesis, collagen/mesh onplants were placed on the CAM and visualised after 3 days (Figure 4). As a positive control for angiogenesis, collagen plugs containing human fibrosarcoma (HT1080) cells were used. Significantly more newly formed blood vessels were observed in the HT1080 plugs compared with plain collagen plugs, both in macrophage-depleted (* *p* = 0.0033) and control embryos (** *p* < 0.001) (Figure 5).

A significantly lower number of invading blood vessels was observed in embryos injected with clodronate liposomes compared with control embryos injected with PBS liposomes (Figure 5). This difference in angiogenesis was observed for both plain collagen plugs (*** *p* < 0.0001) and HT1080 plugs (** *p* < 0.001) (Figure 5). The median angiogenic response in plain collagen plugs was 4.69% for control embryos and 0% for macrophage-depleted embryos; in HT1080 plugs, these percentages were 13.21% and 3.94% respectively (Appendix A). Injection with free clodronate versus injection with PBS did not have a significant effect on angiogenesis, neither in plain collagen plugs (*p* = 0.16) nor in HT1080 plugs (*p* = 0.97) (Appendix A).

## 4. Discussion

To investigate the influence of endogenous macrophages on angiogenesis in the chick CAM model, a successful depletion of these phagocytes is required. Here, we demonstrate that injection with clodronate liposomes effectively reduces macrophage levels in the chick CAM model and that this depletion directly affects vascularisation of collagen onplants. Hence, the chick CAM assay provides a promising model to investigate the role of macrophages in angiogenesis.

Macrophages fulfil several roles during embryonic development, including removal of apoptotic cells and vascular remodelling. They are particularly abundant in regions with increased programmed cell death, such as the inter-digit regions of limbs, the eye primordium and the central nervous system [18]. Experimental reduction of macrophages during murine development leads to developmental issues or death [26,27]. The observation of a lower survival rate in embryos injected with clodronate liposomes corroborates the importance of macrophages during development. The decreased viability was due to macrophage depletion and not to the toxicity of clodronate, as injection with free clodronate did not result in a significantly lower survival rate. Even though injection with clodronate liposomes decreased embryo viability, the majority of embryos did survive, which is most likely due to the fact that injection happened after 8.5 days of incubation, when most crucial developmental stages have already passed [28]. Furthermore, our results show that injection of clodronate liposomes does not entirely deplete the local macrophage population but causes a three- to fourfold reduction. Thus, injection of clodronate liposomes to investigate the effect of macrophage depletion in the CAM model is a viable technique.

Clodronate liposomes have been widely used to deplete macrophages, especially in mice, but also in other species such as dogs, adult chickens and zebrafish [15,29,30,31,32]. It was therefore not surprising to observe that injection with clodronate liposomes effectively reduced the macrophage population in chicken embryos. This opens up possibilities for using the chicken embryo as model to investigate macrophage function in vivo while adhering to the 3R principle. Currently, macrophages are mostly researched in mice and zebrafish, but as these are considered lab animals, use of these models is subjected to the appropriate legislation [23,33]. As a means for depleting macrophages, treatment with clodronate liposomes is straightforward and relatively easy, as it does not require targeted knockdown of specific genes, which is also far less developed in chicken models in comparison with mice [18,34,35,36]. An interesting alternative for depleting macrophages, however, is the use of antibodies to block the action of macrophage colony stimulating factor (CSF1), a growth factor that promotes the differentiation and survival of macrophages [37,38]. This antibody has long been available for mice but an antibody that specifically targets and depletes chicken CSF1 was developed only recently [39]. The two methods differ in their mechanism, as anti-CSF1 antibodies block the differentiation of macrophages early on, while clodronate liposomes target phagocytising macrophages. However, similar results on vascularisation of tumours have been reported [21,22,37,40]. As injection of clodronate liposomes leads to a reduced angiogenic response in the chick CAM model, it would be interesting to determine whether anti-CSF1 antibodies have the same effect.

A point of discussion is whether clodronate liposomes may be ingested by other phagocytising cell types as well, such as heterophils, the myeloperoxidase-lacking avian counterpart of mammalian neutrophils. Several papers have already reported that injection with clodronate liposomes does not affect mammalian neutrophil numbers [23,41,42,43]. Here, we confirm that liposomes injected intravenously in the chick CAM model are mainly taken up by macrophages.

Injection of clodronate liposomes significantly decreased the ingrowth of blood vessels in the collagen onplants, both in plain collagen onplants as well as in onplants containing pro-angiogenic cells. Embryonic macrophages play a role in developmental angiogenesis in mice and zebrafish, and have been demonstrated to express pro-angiogenic markers [15,16,44]. Our current study shows that embryonic chicken macrophages are also involved in blood vessel development in the chick CAM model.

Macrophages can regulate angiogenesis via diverse mechanisms, such as secretion of pro-angiogenic growth factors or matrix remodelling enzymes, or physical interaction with blood vessels [1,3,15]. Earlier studies with the chick CAM assay have indicated that matrix remodelling by heterophils and monocytes/macrophages is important for angiogenesis in this model. Heterophils are the first innate immune cells to invade collagen onplants and secrete matrix metalloproteinase 9 (MMP9) [24]. This allows the subsequent influx of macrophages, which, in turn, secrete another MMP i.e., collagenase (MMP13) [17]. Our current study corroborates these findings and confirms the importance of macrophages for de novo vascularisation of collagen onplants on the CAM. Although heterophil-secreted MMP9 is crucial for angiogenesis [24], our results suggest that heterophils alone are not sufficient for vascular ingrowth in the collagen onplants.

Depletion of macrophages resulted in a lower angiogenic response in both plain collagen plugs and HT1080 plugs. HT1080 cells are highly angiogenic cells often used as a model to induce tumour angiogenesis. They have been shown to induce neovascularisation in the CAM model at levels comparable with maximal growth factor-induced angiogenesis [24]. Their capacity to induce blood vessel growth is related to the secretion of pro-angiogenic factors such as VEGF and angiogenin [45]. Despite the presence of a pro-angiogenic environment in the HT1080 plugs, blood vessel growth in those plugs was also significantly impaired after macrophage depletion, further implying that embryonic macrophages are essential for angiogenesis in the chick CAM model.

It could be debated whether the observed angiogenic effect was due to the depletion of macrophages or due to the clodronate itself. Ribatti and colleagues added free clodronate to gelatin sponges and placed them on the CAM, after which they observed that clodronate inhibited FGF2-induced angiogenesis, likely by the direct influence of topically added highly concentrated clodronate on endothelial cells [46]. However, others report that clodronate only has a marginal effect on endothelial cells while liposome-encapsulated clodronate does not affect them at all [47,48]. Furthermore, free clodronate molecules should not easily enter cells because of their difficult passage through the cell membrane and their short half-life of around 15 min [41]. This makes it unlikely that circulating clodronate would have a strong effect on endothelial cells. Nevertheless, to determine whether the decrease in blood vessel ingrowth in collagen plugs could be due to the effect of clodronate after release from its liposomal encapsulation, free clodronate was injected at the same concentration as the clodronate liposomes. We did not observe a significant difference in angiogenesis after injection with free clodronate, strengthening the hypothesis that the lowered angiogenic response after injection with clodronate liposomes mainly results from the depletion of macrophages. Whether this effect is related to MMP13 secretion by macrophages or through direct interactions with ingrowing capillary sprouts remains to be investigated. Furthermore, other pro-angiogenic factors which can be expressed by macrophages such as VEGF or FGF2 could be investigated in future studies as well.

## 5. Conclusions

Depletion of macrophages in the chick CAM assay can be achieved by injection of clodronate liposomes. This depletion has a direct effect on the vascularisation of collagen plugs, implying that chicken embryonic macrophages play an important role in angiogenesis. The chick CAM assay therefore presents a promising model to investigate the function of macrophages in angiogenesis and related fields of study such as cancer research.

## Figures and Tables

**Figure 1 cells-10-00005-f001:**
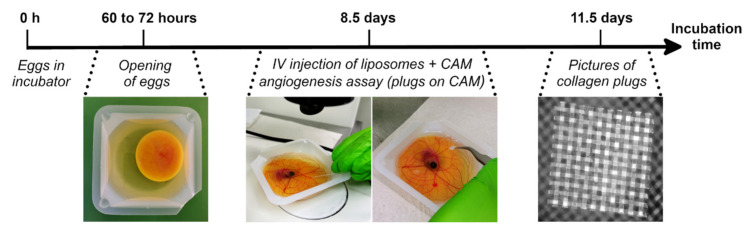
Timeline of the ex ovo chorioallantoic membrane (CAM) angiogenesis assay. At 60 to 72 h of incubation, embryos were removed from their shell and transferred to square weighing boats. At Day 8.5, liposomes containing clodronate or phosphate-buffered saline (PBS) were injected intravenously in the CAM, followed immediately by placement of collagen/mesh onplants on the CAM. After an additional 3 days of incubation, images were taken of the collagen plugs.

**Figure 2 cells-10-00005-f002:**
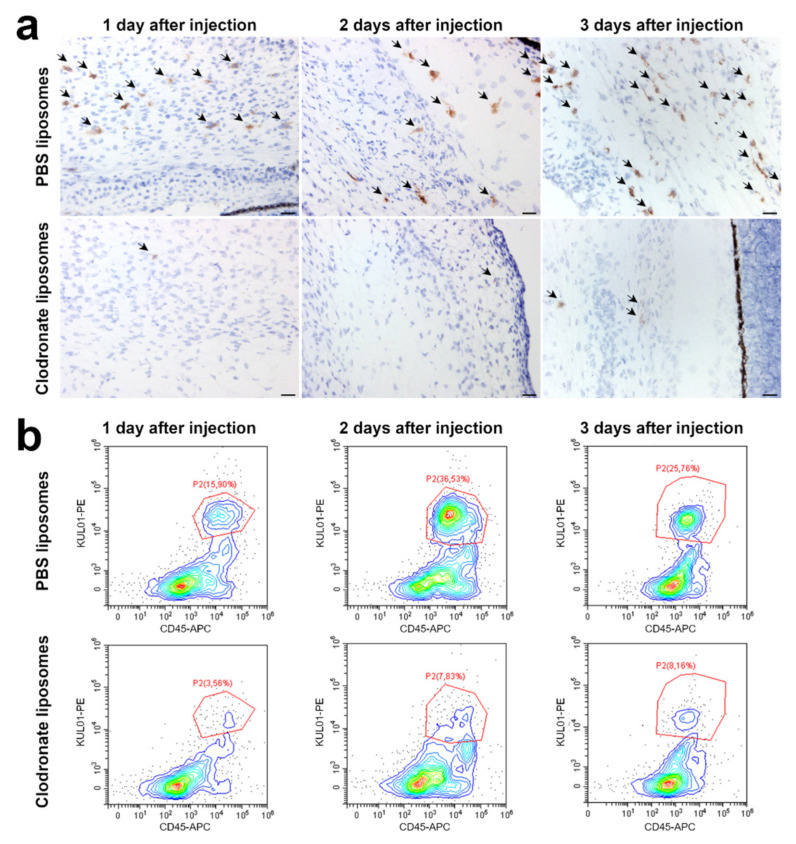
Injection of embryos with clodronate liposomes leads to a depletion of macrophages. (**a**) Immunohistochemistry shows fewer KUL01-positive cells (arrows) in clodronate-injected embryos in the developing choroid and the mesenchymal tissue surrounding the eye sclera compared with chicken embryos injected with PBS liposomes. Macrophage depletion was already visible 24 h after injection of clodronate liposomes and was still visible 3 days after injection. Scalebar = 20 µm. (**b**) Flow cytometry confirmed that injection with clodronate liposomes depletes the macrophage population. KUL01-positive macrophages decrease from 15–37% of CD45-positive leucocytes in control embryos injected with PBS liposomes to 3–9% in clodronate liposome-injected embryos.

**Figure 3 cells-10-00005-f003:**
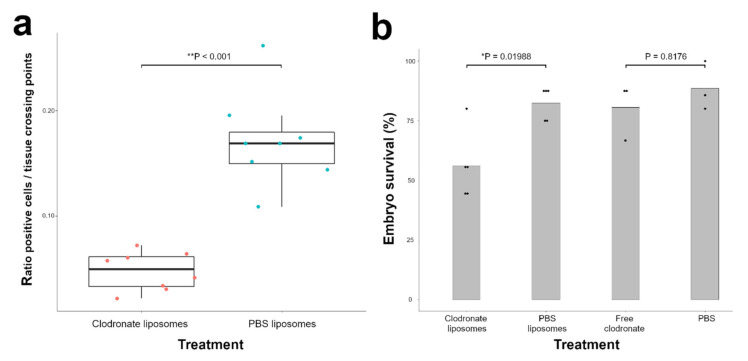
Chicken macrophages are significantly reduced 3 days after injection with clodronate liposomes in the chicken embryo. (**a**) Quantification of KUL01-positive cells by immunohistochemistry revealed a 3.4-fold decrease in KUL01-positive cells (** *p* < 0.001). The orange and blue dots represent the average ratio positive cells/tissue crossing points per individual sample of clodronate liposome and PBS liposome-treated embryos respectively. The Wilcoxon rank sum test was used to analyse statistical significance. Data are presented as boxplots showing the median, first and third quartiles. (**b**) Injection of clodronate liposomes but not free clodronate significantly reduces viability in the chicken embryo: 56% of clodronate liposome-injected embryos were alive 3 days after injection versus 82.5% of PBS liposome-injected embryos (* *p* = 0.01988). On the other hand, embryo viability after injection with free clodronate measured 80.56% versus 88.57% after PBS injection (*p* = 0.8176). The black dots represent the embryo survival % per individual experiment. Statistical significance was determined with the Cochran–Mantel–Haenszel Test.

**Figure 4 cells-10-00005-f004:**
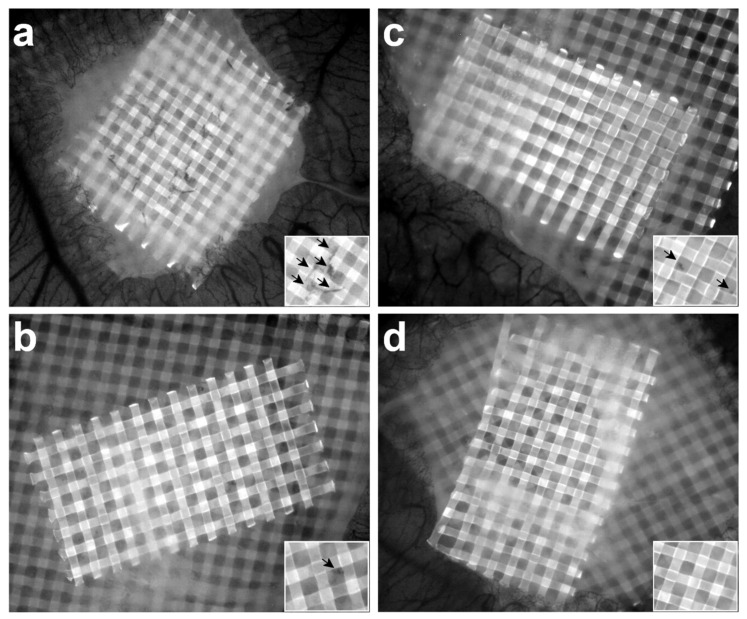
Collagen/mesh onplants were placed on the CAM and visualised after 3 days to allow quantification of angiogenesis. Blood vessels growing above the plane of the top mesh in the collagen plug can be seen as black structures (arrows in insert) against a grey mesh on images taken in the DAPI channel (excitation, 358 nm; emission, 463 nm). Representative images of angiogenic HT1080 plugs of control embryos (**a**) and macrophage-depleted embryos (**b**) and representative images of plain collagen plugs of control embryos (**c**) and macrophage-depleted embryos (**d**). Fewer blood vessels were observed in the plugs of macrophage-depleted embryos than in those of control embryos, both in the HT1080 plugs ((**b**) versus (**a**)) and the plain collagen plugs ((**d**) versus (**c**)).

**Figure 5 cells-10-00005-f005:**
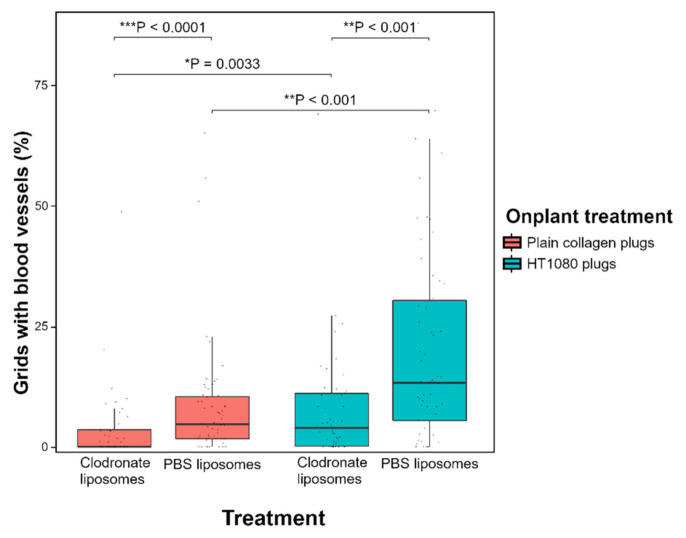
A lower angiogenic response was observed in collagen plugs on the CAM of macrophage-depleted embryos compared with control embryos. Statistical analysis using the Wilcoxon rank sum test revealed a significantly lower amount of ingrowing capillaries in embryos treated with clodronate liposomes, both in plain collagen plugs (*** *p* < 0.0001) and in collagen plugs containing pro-angiogenic HT1080 cells (** *p* < 0.001). The median angiogenic response in plain collagen plugs was 4.69% for control embryos and 0% for macrophage-depleted embryos; in HT1080 plugs, the response was 13.21% and 3.94%, respectively. Furthermore, HT1080 plugs were confirmed to induce more angiogenesis than plain collagen plugs in both macrophage-depleted embryos (* *p* = 0.0033) and control embryos (** *p* < 0.001). Data are presented as boxplots showing the median, first and third quartiles.

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
