# Peer review of "Depletion of Embryonic Macrophages Leads to a Reduction in Angiogenesis in the Ex Ovo Chick Chorioallantoic Membrane Assay"

_cells, 2020, doi:10.3390/cells10010005_

Round 1
Reviewer 1 Report
The study is well-designed and the conclusion reached by the authors are perfectly supported by the obtained experimental data. The results discussion is well-documented and very interesting to read. I just have some minor points:
- Some parts of Figure 3b are missing. In addition, it would interesting to add the cytotoxicity data of free clodronate and PBS. Or add a figure with all cytotoxicity data to the SI.
- What is the size of the liposomes? Are both clodronate and PBS liposomes similar in size? It is known that different liposomes sizes can lead to differences in their biodistribution and therefore in their cellular uptake and in vivo activities. It would be nice to add this data in the materials and methods section.
- Does the angiogenesis study necessarily requires the use of collagen/mesh onplants? Couldn't angiogenesis be directly measure on the CAM? If not, could this be better explained? The authors justify the use of this method on the basis of a paper published by Zijlstra (reference 24).
Reviewer 2 Report
The present research article entitled “Depletion of embryonic macrophages leads to a reduction in angiogenesis in the ex ovo chick chorioallantoic membrane assay” is an interesting work for researchers working in cancer or tumor growth related research. Authors highlighted the role of macrophages in angiogenesis. With CAM model, authors injected clodronate liposome where they found it may led to a significant 3.4 fold reduction of macrophages compared to control. Authors have performed several important endpoints i.e. Ex ovo CAM assay, immunohistochemistry, FACS to confirm the clodronate liposomes depletes the macrophage population. The presentation of paper is quite ok. This research would be of great interest for researcher working on cancer related issues.
Following suggestions could improve the manuscript:
- English is good but still some large sentences could be rephrased for better understanding. Specifically, there is the tendency of abusing of colloquial that in my opinion are not grammatically correct. I would invite the authors to generally revise the language by a professional English editing service to improve the comprehension of the contents. Few examples in abstract are- “chick CAM assay” it should be CAM assay as C stand chick; in vivo should be italic in vivo
- Central message in the abstract is missing. This should be more precise and focused.
- In the introduction section some relevant information is missing. I will suggest to add several recent findings in this area of research to support your data.
- Authors should add a figure in the manuscript summarizing the whole study, including mode of action of clodronate liposomesin depletion of embryonic macrophages in angiogenesis along with mechanism and This will definitely improve the paper quality and easy for readers to understand the research.
- Authors have performed several parameters representing role of macrophages in angiogenesis after injecting with clodronate in CAM model. Several important findings have been presented in the manuscript. Although, in my opinion some additional study will enhance the quality of the manuscript. I will suggest authors to perform couple of oxidative stress i.e. ROS and cytotoxic (DHE assay) related parameters, if possible.
- In figure 2A, indicate all the changes through arrows.
- I did not see any information about the samples size taken for assay and in statistical data. Was this study repeated? Authors should include sample size in the figure(s) captions by defining the statistical methods (Mean ±SD or mean±SEM).
- Author should include the conclusions in the manuscript. If possible, future perspectives of CAM model in the study of cancer and associated diseases will be useful. It should preferably be stated clearly and in a concise manner based on the present study.
Round 2
Reviewer 2 Report
Thank you for the answers on the comments. I am satisfied with authors answers.